Miocene sponge assemblages in the face of the Messinian Salinity Crisis—new data from the Atlanto-Mediterranean seaway

http://orcid.org/0000-0002-5282-5341 Łukowiak Magdalena 1 mlukowiak@twarda.pan.pl
Meiro Gerardo 2
Peña Beltrán 3
Villanueva Guimerans Perfecto 3
http://orcid.org/0000-0002-8433-6585 Corbí Hugo 4
1 Department of Environmental Paleobiology, Institute of Paleobiology, Polish Academy of Sciences , Warszawa, Mazowieckie , Poland
2 Madrid , Spain
3 Jerez de la Frontera, Cádiz , Spain
4 Department of Earth Sciences and the Environment, Universidad de Alicante , Alicante , Spain
Ereskovsky Alexander
Electronic publication date: 2023 Nov 16
Publication date: 2023
Volume: 11
Electronic Location ID: e16277
Received 2023 Jul 12; Accepted 2023 Sep 20
Copyright: © 2023 Łukowiak et al.
Copyright year: 2023
Copyright holder: Łukowiak et al.
License: This is an open access article distributed under the terms of the Creative Commons Attribution License, which permits unrestricted use, distribution, reproduction and adaptation in any medium and for any purpose provided that it is properly attributed. For attribution, the original author(s), title, publication source (PeerJ) and either DOI or URL of the article must be cited.
License URL: https://creativecommons.org/licenses/by/4.0/

Keywords: Messinian, Guadalquivir Basin, Porifera, Tortonian, Strait of Gibraltar

Funding: Spanish Ministry of Science and Innovation/State Research Agency of Spain (AEI) EVAMED (PID2020-118999GB-I00) This work was supported by EVAMED (PID2020-118999GB-I00) funded by the Spanish Ministry of Science and Innovation/State Research Agency of Spain (AEI). The funders had no role in study design, data collection and analysis, decision to publish, or preparation of the manuscript.

==============================
The Messinian Salinity Crisis is considered as one of the most influential Cenozoic events that impacted negatively on the benthic fauna of the Mediterranean area. Changing environmental conditions, including a sharp reduction of water exchange between the Mediterranean Sea and the Atlantic Ocean, altered the geographical ranges of many organisms, including sponges (Porifera). Here, we report a unique assemblage of isolated sponge spicules from the upper Miocene of southwestern Spain. The newly recognized sponge fauna was inhabiting the Guadalquivir Basin—the corridor between the Mediterranean and the Atlantic Ocean at that time. It represents a taxonomically rich sponge community that consisted of members of “soft” and “lithistid” demosponges and hexactinellids. Demosponges are represented by at least thirty-four taxa, while hexactinellids are significantly rarer; only six taxa have been identified. From among eighteen taxa recognized to the species level, at least eight seem to be inhabiting this area to these days; six are recorded from adjacent areas, such as the Western Mediterranean, South European Atlantic Shelf, and the Azores, and three are present in the Red Sea and/or the Northern Atlantic. Intriguingly, some taxa seem to have their closest relatives in distant areas, such as the Indo-Pacific and Japanese waters which suggests that the range of some once widely-distributed populations shrunk after the isolation of the Mediterranean and the Messinian Salinity Crisis, surviving to the present day only in refugia.

Introduction

Sponges (Porifera) are a diverse clade of sessile animals adapted to an aquatic lifestyle whose fossil record reaches at least the lowermost Cambrian (535 million years ago; Antcliffe, Callow & Brasier, 2014); as such, they are among the earliest-diverging multicellular organisms (Hamdi, Brasier & Jiang, 1989). Despite that sponges have often represented dominant components in marine environments, their record is substantially incomplete and the extinct communities are still poorly known. This is true especially for taxa that are being reported through their disassociated skeletal elements—the spicules, which are often the only evidence of their existence. The fossil record of sponges is biased towards the groups with skeletons made of fused spicule nets or spicules connected by articulation that tend to be preserved through their whole bodies—the hexactinellids (Hexactinellida) and the so called “lithistids” (“Lithistida”) which is an informal polyphyletic grouping within the demosponges (Demospongiae). The sponge communities reconstructed based on assemblages of loose spicules are comparatively less frequent. However, the group that is dominant in modern seas—the demosponges—tends to preserve as dissociated spicules (Łukowiak et al., 2022a). This highlights the necessity to analyze loose spicules while reconstructing past sponge communities (see also Łukowiak, 2020).

Sponge faunas from the Miocene of Paratethys and Tethys that are known based on dissociated spicules are quite rare (Alexandrowicz & Tomaś, 1975; Riha, 1982, 1983; Hurcewicz, 1991; Ivanova & Olshtynska, 2004; Pisera & Hladilová, 2003; Matteucci & Russo, 2012; Łukowiak, Pisera & Schlögl, 2014) and the spicule assemblages are usually scarce in terms of the number of spicule types. Little has also been known about sponges from the Paleo-Mediterranean (e.g., Pisera, Cachao & da Silva, 2006; Costa et al., 2021; Hoffmann et al., 2020) and some other areas (Bukry, 1979; Palmer, 1988).

Here, we describe a rich and diverse assemblage of sponge spicules from the Miocene (Tortonian-Messinian) of southern Spain (Guadalquivir Basin). This new material provides a unique snapshot of a sponge community thriving before the onset of the Messinian Salinity Crisis (MSC) which strongly affected the faunal composition of the Mediterranean Sea and changed the water exchange between the Atlantic and Indian Oceans (Hou & Li, 2018 and literature cited therein). As such, it sheds light on how the sponge fauna of this region has been affected by the MSC and the isolation of the Mediterranean Sea (Roveri et al., 2014; Corbí et al., 2016).

Materials and Methods

Collection and processing of the material

Sediment samples were collected in 2020 and 2022, in Cerro Viejo at Jerez de la Frontera (36°42′49.8″N 6°10′25.3″W (36.713822, −6.173706)), and Martín Miguel at Sanlucar de Barrameda (36°47′01.6″N 6°19′13.4″W (36.783784, −6.320380)) (Fig. 1). The isolated siliceous sponge spicules were retrieved from Miocene sediments. Approximately ten sediment samples of variable volume (100 grams) were taken for spicule analysis. Samples were first sieved and macerated in the laboratory using hydrogen peroxide (H2O2, 30%) for 10 h at 90 °C to remove organic matter and to cleanse and separate spicules. Subsequently, the spicules were handpicked under a transmitted light microscope (CX43; Olympus, Barcelona, Spain) and a binocular microscope (SMZ800; Nikon, Warsaw, Poland). Representative spicule types were hand-mounted on scanning electron microscope stubs, coated by sputtering with gold and/or platinum, and photographed using a FEI-Phillips XL-30 SEM at the Institute of Paleobiology (PAS), Warsaw and FEI-Phillips XL-30 SEM in Spain. The spicules are housed in the Museo de Paleontología Universidad de Huelva, Spain, with numbers AMUHU-CE-100.

Figure 1 General geology of the Betic Cordillera and adjacent offshore domains with the distribution of Neogene-Quaternary basins (from Driussi et al., 2015 and Ochoa et al., 2015).

Towards the west the Guadalquivir basin and the position of the studied outcrops (MM: Martín Miguel, CV: Cerro Viejo). Figure modified from García-Veigas et al. (2020).

Taxonomic assignment of loose sponge spicules

Throughout the Cenozoic, sponges maintained body plans and spicule morphologies that are broadly similar to, and often indistinguishable from, those of modern representatives of Porifera, prompting researchers to assign spicules obtained from numerous Cenozoic deposits to lineages (including species-level taxa) that are still present in modern aquatic environments (e.g., Costa et al., 2021; Łukowiak, 2015; Łukowiak et al., 2022b). Owing to the slow evolutionary rates of sponges and the conservative morphology of their skeletons, that can partially explain the longevity of some of their lineages (e.g., Jochum et al., 2012), we have included modern sponge taxa in our comparisons, focusing especially on those that are still present in the region. If no match was found, the studied spicules were compared with those that are present in taxa inhabiting distant geographical locations.

Geological background

The studied area is situated within the Guadalquivir Basin—the foreland basin of the Betic Cordillera located in South of Spain (Fig. 1). The Guadalquivir Basin is an important depression, elongated in NE-SW direction. The basinal sedimentary infill is limited to the south by the External Zone (or South Iberian Paleomargin) of the Betic Cordillera and to the north by the mountains of Sierra Morena, formed by Paleozoic and Mesozoic units of the Iberian Massif (Sanz de Galdeano & Vera, 1991, 1992). The Guadalquivir Basin formed as a foreland basin at the beginning of the late Miocene when the paleogeography of the Cordillera changed drastically once the westward drift of the Internal Zones ceased and the North Betic strait was partially closed (Vera, 2000). The stratigraphical architecture of the Neogene-Quaternary record of this basin has been defined from borehole and seismic data (e.g., Martínez del Olmo et al., 1984; Riaza & Martínez del Olmo, 1996; Sierro et al., 1992, 1996; González-Delgado et al., 2004; Martínez del Olmo & Martín, 2016), although outcrop studies centered mainly on sedimentological aspects are also available (e.g., Molina et al., 1998; García-García et al., 2003; Martínez del Olmo & Díaz, 2004; García et al., 2014; Aguirre et al., 2022). Transverse seismic sections show higher thickness and tectonized sedimentary units in the active margin or southern margin in comparison to the outcrop units of the passive or northern margin. A synthesis of the sedimentary units recorded in this basin can be found in González-Delgado et al. (2004), Sierro et al. (1996), and Martínez del Olmo & Martín (2016).

The dissociated sponge spicules originate from a Miocene unit known as moronitas or albarizas (a colloquial Spanish term referred to the white color of the strata) from the locality Jerez de la Frontera (Calderón & Paul, 1886). The typical facies of moronitas are laminated white (slightly yellow) marls, also called diatomaceous marls with a rich assemblage of calcareous (nanoplankton and Foraminifera) and siliceous (diatoms, radiolarians, silicoflagellates, and spicules of siliceous sponges) microfossils. From the biostratigraphical perspective, several authors assigned the moronitas to the Eocene, Oligocene, and Miocene (see the review in Colom & Gamundi, 1951), although the biostratigraphical data offered by Bustillo & López García (1995) establish three different episodes in the middle and late Miocene based on the diatom biozones. Diatoms and calcareous nannoplankton, as well as microfauna (radiolaria and foraminifera) are characteristic for a relatively shallow open marine basin (200–300 m) with a high organic production (IGME, 1988). The sedimentation with high diatomaceous content was the consequence of formation in a high-biogenic productivity area caused by coastal upwelling (Bustillo & García, 1997; López García, 1995).

The age and the paleoenvironment of the studied deposits (Martín Miguel and Cerro Viejo) was estimated based on the foraminiferids analysis. Each sample was wet-sieved to collect the >125 μm fraction, determining the presence of biostratigraphical marker species, the percentage of planktonic Foraminifera and the most representative benthic Foraminifera groups. The ratio of planktonic-to-benthic foraminiferids close to 30%, and the fact that benthic foraminiferids were dominated by cassidulinids, discorbids, buliminids, lagenids, and uvigerinids, suggests that the assemblages were inhabiting the outer shelf environment. The biostratigraphical analysis (the presence of Neogloboquadrina acostaensis) allowed establishing a Tortonian-Messinian (late Miocene) age for the studied sites.

Results

Systematic paleontology

Class Demospongiae Sollas, 1885

Subclass Heteroscleromorpha Cárdenas, Pérez & Boury-Esnault, 2012

Order Axinellida Lévi, 1953

Family Raspailiidae Nardo, 1833

Genus Plocamione Topsent, 1927

Type species. Plocamione dirrhopalina Topsent, 1927 (type by monotypy).

Plocamione cf. dirrhopalina Topsent, 1927

Figures 2A and 2B

Figure 2 Sponge spicules of the late Miocene of the Guadalquivir Basin (Demospongiae).

(A and B) Acanthostyles of Plocamione cf. dirrhopalina; (C) styles of Rhabderemia sp.; (D) spiral strongyle of Spiroxya sp.; (E) wavy diactine; (F) wavy diactine of Bubaris subtyla; (G) vermicular diactine of Monocrepidium vermiculatum; (H) acanthoxea of Histodermella cf. ingolfi; (I) diplaster of Diplastrella sp.; (J) selenaster of Placospongia decorticans (transmitted light); (K and L) diancistras of Hamacantha (H.) johnsoni; (M) diancistra of ?Hamacantha (Hamacantha) lundbecki; (N and O) spicules of Crambe cf. tuberosa; (P and Q) diancistra of ?Hamacantha (Vomerula) papillata; (R) sigma of unknown sponge; (S and T) spicules of Discorhabdella tuberosocapitata; (U) acanthotylostyle of Discorhabdella sp.; (V and W) aciculospinorhabds of Latrunculiidae indet. 1; (X) aciculorhabd of Latrunculiidae indet. 2; (Y) spiny strongyle of Sceptrella cf. biannulata; (Z and AA) anisochelae of Mycale (Mycale) grandis; (BB) placochela of Euchelipluma pristina; (CC and DD) club-shaped spicules of M. (Rhaphidotheca) marshallhalli.

Material. Two spicules.

Description. The spicules are stout curved acanthostyles. They are 120 and 150 µm long and possess ornamented heads on both spicule ends; in one case the heads are well-defined and the spicule is equipped with additional ornamented swelling in the spicule center. In the second case, only one head is well-developed and the other head smoothly passes into a median ornamented swelling which is less pronounced.

Remarks. These spicules are of almost identical morphology as spicules of modern axinellid Plocamione, especially P. dirrhopalina (compare with Ridley & Duncan, 1881, pl. 23.19). This species is known today from the Mediterranean and the Azores (de Voogd et al., 2023) and is recorded from considerable depths of about 1,200 m (Topsent, 1928).

Order Biemnida Morrow et al., 2013

Family Rhabderemiidae Topsent, 1928

Genus Rhabderemia Topsent, 1890

Type species. Rhabderemia pusilla (Carter, 1876) (type by subsequent designation).

Rhabderemia sp.

Figure 2C

Material. Single spicule.

Description. Curved, 270 µm long style. The spicule’s upper part is delicately ornamented with minute spines and strongly curved in a manner of a walking stick. The spicule thickness decreases gradually till the other end.

Remarks. The studied curved style strongly resembles those of Rhabderemia fascicularis Topsent, 1927 (Topsent, 1928, pl. X, fig. 25). However, the ornamentation of the modern species is of different character and does not cover the curved part of the spicule. This species is noted today from the N Atlantic (Azores, Canaries, and Madeira; de Voogd et al., 2023). There are also other species of Rhabderemia, e.g., R. typica or R. profunda which are characterized by presence of rhabdostyles in their skeletons but they usually are much smaller, or they are differently ornamented and curved. Likewise, the rhabdostyles may appear in some poecilosclerid and halichondrid species, but in this case assignment of the studied spicule to some unidentified species of Rhabderemia seem to be the most plausible.

Order Bubarida Morrow & Cárdenas, 2015

Family Bubaridae Topsent, 1894

Genus Bubaris Gray, 1867

Type species. Bubaris vermiculata (Bowerbank, 1866) (type by original designation).

Bubaris subtyla Pulitzer-Finali, 1983

Figure 2F

Material. Two spicules.

Description. Slender, irregular, sinuous to vermicular 200 and 600 µm long anisoxeas.

Remarks. Such spicule type characterizes sponges of the genus Bubaris. One species is noted from the same area (de Voogd et al., 2023) and possess identical anisoxeas of length 80 to 240 µm (Pulitzer-Finali, 1983, fig. 47). It is Bubaris subtyla and it is recorded from depth of 120–150 m (Pulitzer-Finali, 1983). There is also one bigger anisoxea that could have belonged to other species of Bubaris i.e., B. carcisis which is characterized by presence of longer spicules (600–1,870 µm in length; Vacelet, 1969, p. 36) and is noted from the Mediterranean and the Celtic Sea (de Voogd et al., 2023) but this assignment is tentative. Likewise, there are some other bubarid species that possess similar spicules in their skeletons, e.g., Auletta pedunculata (Topsent, 1896) or even some ancorinids (e.g., Jaspis) that are characterized by sinuous diactines and are recorded from the Mediterranean (de Voogd et al., 2023), so the unequivocal assignment of the longer diactines is not possible.

Genus Monocrepidium Topsent, 1898

Type species. Monocrepidium vermiculatum Topsent, 1898 (type by monotypy).

Monocrepidium vermiculatum Topsent, 1898

Figure 2G

Material. Single spicule.

Description. Stout, vermiculate, tuberculate to annulate strongyle of at least 150 µm of length (seem to be broken at one tip).

Remarks. This strongly vermicular diactine characterizes sponges of the genus Monocrepidium. Among three species of Monocrepidium. One, M. vermiculatum is noted from the Mediterranean Sea (de Voogd et al., 2023). It also has been recorded in the N Atlantic and is restricted to rather deep waters (121–600 m depth; Alvarez & van Soest, 2002).

Order Clionaida Morrow & Cárdenas, 2015

Family Clionaidae d’Orbigny, 1851

Genus Spiroxya Topsent, 1896

Type species. Spiroxya heteroclita Topsent, 1896 (type by monotypy).

Spiroxya sp.

Figure 2D

Material. Single spicule.

Description. Slender, spirally arranged microspined strongyle. This 65 µm long spicule is twisted with two twists and is delicately ornamented on the convex parts of the coils.

Remarks. There are several species of clionaid genus Spiroxya noted from the Mediterranean and the Atlantic areas that are characterized by such spiral strongyles. For example, S. abyssorum (Carter, 1874) is noted from the N Atlantic (Celtic Sea; de Voogd et al., 2023), but their spiral strongyles are thinner and slenderer. On the other hand, S. heteroclita Topsent, 1896 which is noted from the Mediterranean, possess spicules of the same shape and length that the fossil one. Nevertheless, they seem not to be ornamented (compare with Topsent, 1900, fig. 11c). Spicules of Spiroxya corallophila (Calcinai, Cerrano & Bavestrello, 2002) are of similar shape, size and are sometimes microornamented (Calcinai, Cerrano & Bavestrello, 2002). This species is found in the western Mediterranean (de Voogd et al., 2023) and was retrieved from the coral living on the depth of 30–35 m (Calcinai, Cerrano & Bavestrello, 2002). On the other hand, Spiroxya levispira (Topsent, 1898) possess similarly ornament spiral strongyles as well (compare with van Soest & Beglinger, 2009, fig. 3c) and it is noted from 80–700 m of the Mediterranean and N Atlantic Ocean (Rosell & Uriz, 2002; de Voogd et al., 2023). Spicules of shallow water Mediterranean species Spiroxya sarai (Melone, 1965) are also within the size range of the fossil spicule, but are smooth (Corriero, Abbiati & Santangelo, 1997). Likewise, a shallow water Spiroxya spiralis, which is noted today from the Azores and western Caribbean Sea, is characterized by the presence of microspined spiral strongyles of similar shape and size (Rützler et al., 2014). The studied strongyle belongs, most probably, to one of these species.

Family Spirastrellidae Ridley & Dendy, 1886

Genus Diplastrella Topsent, 1918

Type species. Diplastrella bistellata (Schmidt, 1862) (type by original designation).

Diplastrella sp.

Figure 2I

Material. Single spicule.

Description. A microsclere spiraster/diplaster, 50 µm in length.

Remarks. Such characteristic diplasters are found in spirastrellid species Diplastrella. There are three species of Diplastrella noted in the Mediterranean: D. ornata Rützler & Sarà, 1962, D. boeroi Costa et al., 2019, and D. bistellata (Schmidt, 1862). That later one appears also in the South European Atlantic Shelf area which is a recent equivalent of the fossil site (de Voogd et al., 2023). The microscleres of this species are of comparable size and shape (see Rützler, 2002a, fig. 2d). The other spicule types characteristic for this species have not been found in the studied material, however.

Family Placospongiidae Gray, 1867

Genus Placospongia Gray, 1867

Type species. Placospongia melobesioides Gray, 1867 (type by original designation).

Placospongia decorticans (Hanitsch, 1895)

Figure 2J

Material. Single spicule.

Description. Young, bean-shaped spicule, 66 µm in length with surface covered by minute irregular short rays. The rays’ ridges connect with other spines ridges in a polygonal manner.

Remarks. This spicule is identical with selenasters of Placospongia decorticans (compare e.g., with Cárdenas, 2020, fig. 7f)—a species that is noted from all over the Mediterranean Sea and the E Atlantic Ocean, including South European Atlantic Shelf (de Voogd et al., 2023). Sponges of this species are shallow-water inhabitants.

Order Merliida Vacelet, 1979

Family Hamacanthidae Gray, 1872

Genus Hamacantha Gray, 1867

Type species. Hamacantha (Hamacantha) johnsoni (Bowerbank, 1864) (type by original designation).

?Hamacantha (Hamacantha) lundbecki Topsent, 1904

Figure 2M

Material. Single spicule.

Description. Stout, 140 µm-long diancistra. This microsclere possess hook-shaped ends, sometimes notched at the point where they join the shaft. The shaft is straight, with a narrowing on the middle and the ends not at the same plane as the shaft (ca. 45 degrees in opposing directions).

Remarks. This spicule is almost identical with diancistras of two Hamacantha species, namely H. (Hamacantha) hortae and H. (H) lundbecki. The diancistras of both these species possess the characteristic features e.g., stout, straight shaft with a subtle depression in the middle, with recurved hook-like fimbriae on both sides of the shaft, and ends usually bent in opposing directions (Topsent, 1904, pl. XVI, fig. 7b; Santín et al., 2021, fig. 6d). They also both are characterized by very similar size (H. hortae: 123–139 μm and H. (H) lundbecki: 145–155 μm; Santín et al., 2021 and Topsent, 1928, respectively). The first species lives on the 600 m of depth in W Mediterranean and the other one is noted from the studied area, namely South European Atlantic Shelf (de Voogd et al., 2023). Due to the presence of H. (H) lundbecki in the studied area, the analyzed spicule belongs most probably to this species.

?Hamacantha (Vomerula) papillata Vosmaer, 1885

Figures 2P and 2Q

Material. Three spicules.

Description. Long (200–270 µm), slender diancistras with shaft thinning from the distal parts to the center. The spicule shafts are equipped with wing-like structures called fimbriae (compare with Boury-Esnault, Pansini & Uriz, 1994, fig. 72c). One of the spicules is characterized by the ends not in the same plane as the shaft (Fig. 2P).

Remarks. The size of diancistras suggests that they may belong to one of the two Mediterranean species characterized by big diancistras, namely Hamacantha (Vomerula) megancistra with 220–280-long diancistras (Pulitzer-Finali, 1978), or H. (Vomerula) papillata, with diancistras that can be 220–260 µm long (Boury-Esnault, Pansini & Uriz, 1994). Despite size similar to both these species, the presence of fimbriae on the spicule’s shafts, suggests that they belong to the latter species. Hamacantha (Vomerula) papillata is recorded not only from the Mediterranean, but also from South European Atlantic Shelf from the depth range 185 to 1,600 m (de Voogd et al., 2023).

Hamacantha (Hamacantha) johnsoni Bowerbank (1864)

Figures 2K and 2L

Material. Two spicules.

Description. Short, 150–170 µm long, stout diancistras with a blade-like surfaces on spicule ends and along the shaft (Fig. 2L).

Remarks. These spicules may belong to Hamacantha (Hamacantha) johnsoni which is characterized by diancistras that are 113 to 167 µm long and are characterized by thin blade-like surfaces on the spicule ends (Boury-Esnault, Pansini & Uriz, 1994, fig. 73b). This species is recorded from the Mediterranean, including Alboran Sea and the South European Atlantic Shelf (Hajdu, 2002; de Voogd et al., 2023), and inhabits deep water of 170 to 924 m of depth (Boury-Esnault, Pansini & Uriz, 1994).

Order Poecilosclerida Topsent, 1928

Family Coelosphaeridae Dendy, 1922

Genus Histodermella Lundbeck, 1910

Type species. Histodermella ingolfi Lundbeck, 1910 (type by subsequent designation).

Histodermella cf. ingolfi

Figure 2H

Material. Single spicule.

Description. Heavily spined acanthoxea of a length of 250 μm. The coarse spines are chaotically arranged along the whole spicule length besides the pointed tips. They seem to get bigger to the spicule center and they are curved to the center as well.

Remarks. This kind of megascleres—acanthoxeas with smooth, pointed tips, is recorded in coelosphaerid genus Histodermella. The nearest to the study area species of Histodermella is Histodermella ingolfi, which is noted today from the N Atlantic Ocean (de Voogd et al., 2023). It possesses acanthoxeas with spine-free spicule tips and is of identical size (compare with Stephens, 1921, pl. 3, fig. 4). This species is also characterized by sigmas which can be up to 60 μm long. Histodermella ingolfi was reported living in considerable depths of about 700 to 1,400 m (Stephens, 1921).

Family Crambeidae Lévi, 1963

Genus Crambe Vosmaer, 1880

Type species. Crambe crambe (Schmidt, 1862) (type by original designation).

Crambe cf. tuberosa

Figures 2N and 2O

Material. Three spicules.

Description. Characteristic desmoid spicules with several arms spreading/diverging as a star in a one plane from the ornamented (spined) center. These astrose desmas are up to 150 × 100 µm in diameter.

Remarks. These very characteristic spicules are common in Crambe. There are three species of this genus inhabiting Mediterranean Sea today, Crambe crambe (Schmidt, 1862), C. tuberosa Maldonado & Benito, 1991, and C. tailliezi Vacelet & Boury-Esnault, 1982. The latter two possess characteristic astrose desmas that are almost identical with the fossil ones (see Uriz & Maldonado, 1995); moreover, C. tuberosa is noted from the Alboran Sea and N Atlantic Ocean so probably the spicules belong to this species (de Voogd et al., 2023).

Genus Discorhabdella Dendy, 1924

Type species. Discorhabdella incrustans Dendy, 1924 (type by monotypy).

Discorhabdella tuberosocapitata Dendy, 1924

Figures 2S and 2T

Material. One tylostyle and one acantotylostyle.

Description. A tylostyle fragment is 600 µm long, but the missing tip could have made the spicule at least 100 µm longer. The tylostyle head which is about 25 µm in diameter consist of up to 10 short, mammilliform projections (Fig. 2S). The other spicule is strongly ornamented stout, 170 µm long, acanthotylostyle with well-developed spiny head (Fig. 2T).

Remarks. Both these spicule types characterize crambeid genus Discorhabdella. There are at least three species of Discorhabdella with tylostyles with ornamented heads, namely D. incrustans, D. hindei, and D. tuberosocapitata. Discorhabdella hindei inhabits Mediterranean Sea today. On the other hand, D. tuberosocapitata possess both the tuberculated tylostyles and the echinating acanthotylostyles in its skeleton. Moreover, the tylostyles and acantotylostyles of this species are of comparable size (600 and 130 µm, respectively; according to Boury-Esnault, Pansini & Uriz, 1992). It is recorded form the Azores today from depth of 534–604 m (Boury-Esnault, Pansini & Uriz, 1992; de Voogd et al., 2023).

Discorhabdella sp.

Figure 2U

Material. Single spicule.

Description. Highly ornamented with minute spines, stout acanthotylostyle, 200 µm in length.

Remarks. The spicule resembles pseudoastrose acanthotylotes that are characteristic for some species of Discorhabdella e.g., D. incrustans, D. hindei, and D. tuberosocapitata (Uriz & Maldonado, 1995). From among nine species of Discorhabdella two are known from the Atlanto-Mediterranean region; none of them possesses identical spicules. But spicules of the geographically distant relatives from Japan, e.g., D. hispida or D. misakiensis, possess acanthostyles that resemble those from the fossil material (compare with Ise et al., 2021, fig. 4a and 6a, b, respectively). The spicules might belong or to this species, or to other, unknown so far from this area species of Discorhabdella.

Latrunculiidae indet. 1

Figures 2V and 2W

Material. Two spicules.

Description. Ornamented aciculospinorhabds, about 60 µm in length, with three subsidiary whorls and one apical whorl. In one spicule the subsidiary whorls divide dichotomously (Fig. 2V) and the apical whorl is a solitary, long spine (about 30 µm). The other spicule is characterized by whorls which also divide but the apical whorl is shorter (20 µm) and divides at the middle point on three spines which are directed away from the shaft (Fig. 2W).

Remarks. There is a resemblance of these spicules to spicules of the sponges of the latrunculiid genus Cyclacanthia (Samaai, Govender & Kelly, 2004). Spicules of Cyclacanthia may also be ornamented and with apex with a single long spine and two subsidiary whorls (see e.g., C. bellae Samaai et al., 2003 illustrated in Samaai et al., 2020, fig. 14a). However, none out of four species of this genus is recorded today from the area of N Atlantic or the Mediterranean Sea. These sponges inhabit South African waters (de Voogd et al., 2023). The studied spicules may belong to one of the Latrunculia species as well, e.g., L. (A.) biformis or L. apicalis (compare with Sim-Smith et al., 2022, fig. 62f and 64, respectively). However, all of them inhabit the Southern Hemisphere today (de Voogd et al., 2023).

The aciculospinorhabds characterize also sponges from the family Podospongiidae. Among six species of podospongiids only two are present in this area, namely Podospongia loveni Barboza du Bocage, 1869 (noted from the Mediterranean) and Neopodospongia normani (Stephens, 1915) recorded from the N Atlantic (de Voogd et al., 2023). The studied spicules are similar to those of P. loveni (compare with Cristobo et al., 2009, fig. 5), however, in contrast to spicules of P. loveni, described here spicules are ornamented. In contrast, aciculospinorhabds of N. normani, these spicules possess apical whorl made of three spines, but also one additional central spine and only one collar of whorls instead of two (compare with Stephens, 1915, pl. V, fig. 2b).

Latrunculiidae indet. 2

Figure 2X

Material. Single spicule.

Description. Small, 115-µm-long aciculodiscorhabd. The spicule possesses two whorls (median/subsidiary) that consist of delicately ornamented, diverging dichotomously spines. The apical whorl also consists of a single spine with a small whorl of minute spines in the middle of the spine length. Manubrium comprises five, delicately ornamented, spines.

Remarks. This spicule is similar morphologically to latrunculiid and podospongiid spicules. The latrunculiid genus Tsitsikamma (illustrated e.g., in Li, Kelly & Tasdemir, 2021, fig. 1h) ich characterized by the same features, i.e., subsidiary and median whorls composed of slightly ornamented spines that divide; an apical whorl is made of single spine and a manubrium as a whorl of spines. On the other hand, the spicule resembles also isospinodiscorhabds of latrunculiid Cyclanthia bellae Samaai & Kelly (Samaai, Govender & Kelly, 2004, fig. 2f). Likewise, there are some Sceptrella species whose spicules show some resemblance to studied fossil aciculodiscorhabd. Also, some podospongiid spicules, e.g., of Red Sea inhabitant, Diacarnus ardoukobae Kelly-Borges & Vacelet, 1995 fall within the same morphological plan (compare with e.g., Kelly-Borges & Vacelet, 1995, fig. 3i). However, the assignment to latrunculiids is the most possible.

Poecilosclerida indet.

Material. Two spicules.

Figures 3Q and 3T

Figure 3 Sponge spicules of the late Miocene of the Guadalquivir Basin (Demospongiae).

(A and B) Cladotylotes of Acarnus sp.; (C) polytylote style of ?Acarnus sp.; (D) acanthostrongyle of Antho sp.; (E and F) styles of Clathria sp.; (G–J) acanthostyles of Clathria sp.; (K and L) asters of Tethya sp.; (M) annulate triod of Annulastrella ornata; (N and O) acanthotriaenes of Thrombus abyssi; (P) acanthoxea of Alectona millari; (Q) tylote of Poecilosclerida indet.; (R and S) anatriaenes of Tetillidae indet.; (T) acanthostyle of Poecilosclerida indet.; (U) triod of Astrophorina indet.; (V) calthrop of Astrophorina indet.; (W) broken dichotriaene of Astrophorina indet.; (X) longshafted dichotriaene of Astrophorina indet.; (Y) anatriaene of Astrophorina indet.; (Z–BB) desmas of “lithistida” indet.

Description. There are two spicules that exhibit some poecilosclerid characters. These are, for example, 480-µm-long, slender tylote (Fig. 3Q) and 320-µm-long acanthostyle (Fig. 3T).

Remarks. The tylotes are recorded from the poecilosclerid families Microcionidae, Coelosphaeridae, Tedanidae or Acarnidae. Also, acanthostyles are found in many poecilosclerid groups, e.g., Raspailiidae, Hymedesmiidae, Crellidae, and Microcionidae (Hooper & van Soest, 2002). Due to the general shape of these spicules, they assignment to the lower than poecilosclerid affinity is not possible.

Family Latrunculiidae

Genus Sceptrella

Sceptrella cf. biannulata (Topsent, 1892)

Figure 2Y

Material. Two spicules.

Description. The spicules are stout, 90-µm-long strongyles with not numerous, big spines in two whorls arranged regularly along the spicule. The not well pronounced spicules’ heads are also covered with spines. The spines are not numerous, about 15–20 µm long and are of the same size along the whole length of the spicule.

Remarks. These spicules resemble those of Podospongia loveni Barboza du Bocage, 1869 (compare with Kelly & Samaai, 2002, fig. 1b)—a podospongiid sponge that inhabits today the Mediterranean Sea, including the Alboran Sea area (de Voogd et al., 2023). These strongyles also strongly resemble spicules of Sigmosceptrella quadrilobata Dendy, 1922 illustrated by Dendy (1922; pl. 18, fig. 4b). However, the same species illustrated by Kelly-Borges & Vacelet (1995) seem to have different spicule types (compare with Figs. 3E–3G). Nevertheless, the studied spicules show the biggest similarity to those of Latrunculia biannulata (today Sceptrella biannulata (Topsent, 1892) illustrated by Topsent (1904, pl. XII, fig. 6h).

Likewise, the annulated style assigned here to Acarnus (Fig. 3C), might have also belonged to Sceptrella as very similar spicules are illustrated by Topsent next to strongyles (1904, pl. XII, fig. 6a). Sigmosceptrella quadrilobata is recorded today from the Azores (de Voogd et al., 2023).

Family Mycalidae Lundbeck, 1905

Genus Mycale Gray, 1867

Type species. Mycale (Mycale) lingua (Bowerbank, 1866) (type by subsequent designation).

Mycale (Mycale) cf. grandis Gray, 1867

Figures 2Z and 2AA

Material. Seven spicules.

Description. Anisochelae (140–180 µm in length) with long, sharply pointed upper alae; the lower alae are smaller, broad and rounded; the lower lateral alae are positioned slightly higher than the median ones.

Remarks. The characteristic size and shape of the anisochelae alae displays great resemblance with anisochelae of Mycale (Mycale) grandis (van Soest, Aryasari & De Voogd, 2021, fig. 80c), a species which is noted from the Red Sea, Indian Ocean and Indonesia today (de Voogd et al., 2023). No other spicule types (smaller anisochelae, sigmas or mycalostyles) characteristic for this species had been found in the sample.

Mycale (Rhaphidotheca) marshallhalli (Saville, 1870)

Figures 2CC and 2DD

Material. Four spicules.

Description. One complete exotyle which is 280 µm long and possess round, about 30 µm long head (Fig. 2CC). The other exotyle is incomplete and measures about 270 µm; it is equipped with characteristic, 50-µm-long, pear-shaped head with a delicate ornamentation on its top (Fig. 2DD).

Remarks. Club-shaped spicules are very similar to the spicules occurring in Mycale (Rhaphidotheca) marshallhalli (compare with Stephens, 1921, pl. II, fig. 1b). However, the exotyles of M. (R.) marshallhalli are big: 800–1,400 µm (van Soest & Hajdu, 2002) in contrast to fossil ones. However, the broken exotyle could have reach the size of the modern Mycale spicules when complete. This species is also characterized by presence of big, 250–400-μm-long anisochelae and about 200-μm-long sigmas (Stephens, 1921). Sigmas of comparable size had been found in the studied material as well (Fig. 2R) but dur to their very general shape their definite affinity cannot be established. M. (R.) marshallhalli is noted from the South European Atlantic Shelf and N Atlantic (de Voogd et al., 2023) from waters of about 75 to over 900 m of depth (Stephens, 1921).

Family Cladorhizidae Dendy, 1922

Genus Euchelipluma Topsent, 1909

Type species. Euchelipluma pristina Topsent, 1909 (type by original designation).

Euchelipluma pristina Topsent, 1909

Figure 2BB

Material. Single spicule.

Description. An 85-µm-long and 18 µm-width placochela with delicate ornamentation along the inner part of the spicule alae.

Remarks. This type of spicules characterizes two genera within two poecilosclerid families, Guitarridae and Cladorhizidae (Hajdu & Lerner, 2002; Hajdu & Vacelet, 2002). In Guitarridae only one genus, Guitarra, possess placochelae in its skeleton. This genus has four representatives in the vicinity: G. laplani, G. solorzanoi (both from South European Atlantic Shelf), G. fimbriata (Celtic Sea), and G. voljuta (Azores; de Voogd et al., 2023). However, none of the spicules that characterize these species possess serration on the spicule ridges. In the family Cladorhizidae the placochelae appear only in one species of the genus of Asbestopluma and within the genus Euchelipluma. The latter one contains five species only and only one is recorded from the nearby area of Cape Verde on the Atlantic Ocean (de Voogd et al., 2023). This is Euchelipluma pristina Topsent, 1909 and is characterized by having spicules identical in shape and size with the distinctive serration on the spicule ridges (compare with Desbruyeres, Segonzac & Bright, 2006; fig. 5f).

Family Acarnidae Dendy, 1922

Genus Acarnus Gray, 1867

Type species. Acarnus innominatus Gray, 1867 (type by monotypy).

Acarnus sp.

Figures 3A–3C

Material. Three spicules.

Description. Two 280 and 350 µm long cladotylotes with smooth shafts with one end provided with three hook-like clads, and the second ends strongly ornamented by mammilliform projections (Figs. 3A and 3B). The third spicule is 525 µm long polytylote style with finely spined head (Fig. 3C).

Remarks. The cladotylotes can be assigned to one of Acarnus species. In the Mediterranean Sea there are only two species of Acanus recorded, A. levii, and A. tortilis Topsent, 1892 (former A. polytylus). This second one is also recorded from the South European Atlantic Shelf (de Voogd et al., 2023). However, in both these species, the cladotylote’s distal end is not developed as small mammilliform projections, but as small hooks. Beside this, the spicules’ shaft is covered by rare spines which is not observed in the fossil spicules. The other species noted from the nearby area are A. thielei Lévi, 1958, A. wolffgangi Keller, 1889 (both from the Red Sea), and A. souriei (Lévi, 1952) (from the Azores; de Voogd et al., 2023). The first one possesses similar but smaller cladotylotes (up to 280 µm; van Soest, Hooper & Hiemstra, 1991).

Interestingly, the studied spicules are almost identical with cladotylotes of modern Acarnus claudei van Soest, Hooper & Hiemstra, 1991, but this species occurs recently only in the S Africa (de Voogd et al., 2023). Also A. erithacus which is recorded from western coasts of N America possess cladotylotes with ornamented head (van Soest, Hooper & Hiemstra, 1991).

The other spicule, the polytylote style with spined head, strongly resembles polytylote styles of A. tortillis (former A. polytylus; Pulitzer-Finali, 1983, fig. 70). The delicate spination of the cladotylote head and the character of the rhabds is identical with these spicules. However, the size of the fossil spicule is a little bigger than of the modern Acarnus which is up to 430 µm. The definite assignment of this spicule is problematic, because similar spicules can be also found in few other Mediterranean species, e.g., Polymastia polytylota Vacelet, 1969, Latrunpagoda multirotalis (Topsent, 1927), and Hymedesmia (Stylopus) nigrescens (Topsent, 1925). However, none of these spicules possess spination on the spicule head.

Family Microcionidae Carter, 1875

Genus Antho Gray, 1867

Type species. Antho (Antho) involvens (Schmidt, 1864) (type by original designation).

Antho sp.

Figure 3D

Material. Two spicules.

Description. The 140 and 180-µm-long, slightly curved acanthostrongyles with spined surface. The spines are minute and placed without any order on the spicule surface, but on the faintly developed heads they are more densely arranged than on the rest of the spicule. On the spicule tips, the spines seem to be a bit curved and directed to the spicule center.

Remarks. Similar acanthose strongyles belong to Antho (Antho) dichotoma (Linnaeus, 1767), A. (Acarnia) coriacea (Bowerbank, 1874), A. (Acarnia) signata Topsent, 1904, and A. (Antho) inconstans (Topsent, 1925). The described here spicules resemble the most the acanthostrongyles of the two last species (compare with Topsent, 1904, pl. 14.1, van Soest, Beglinger & de Voogd, 2013, fig. 48b and Topsent, 1925, fig. 15c). All these species are noted from the Mediterranean Sea and the last two were recorded also from the East European Atlantic Shelf (de Voogd et al., 2023). There are some other sponge species as well, that possess similar spicules, e.g., hymedesmiid Plocamionida ambigua (Bowerbank, 1866) or myxillid Ectyonopsis ramosa Carter, 1883, but spicules of these species are equipped with better developed spines (compare with van Soest, 2002, fig. 2g). Still, the assignment of studied spicules to Antho seem the most adequate. However, due to lack of other spicule types that would allow the identification of species, the assignment only to genus level is possible. The other spicule noted in the fossil state, namely the smooth style (Fig. 3E) shows some resemblance to some species of Plocamia and Antho which are also characterized by presence of acanthostrongyles (compare with van Soest, Beglinger & de Voogd, 2013, fig. 46, 47, 48).

Genus Clathria Schmidt, 1862

Type species. Clathria (Clathria) compressa Schmidt, 1862 (type by subsequent designation).

Clathria sp.

Figures 3E–3J

Material. Six spicules.

Description. There are several styles types in the studied material: 580 µm long, smooth and curved ones (Fig. 3E); those that are about 380 µm long and delicately ornamented on the spicule tip (Fig. 3F), 400 µm long with ornamented upper half of a spicule (Fig. 3G), and three, long and slender acanthostyles (680, 550, and 620 µm long; Figs. 3H–3J, respectively) with strongly ornamented heads.

Remarks. These types of spicules are noted in the microcionid genus Clathria. The smooth stout style (Fig. 3E) shows great resemblance to spicules of Clathria (Clathria) hjorti (Arnesen, 1920) which is recorded from the N Atlantic Ocean (de Voogd et al., 2023), C. (C.) coralloides, noted from the Adriatic Sea, and C. (C.) arecifera noted from the Azores (compare with van Soest, Beglinger & de Voogd, 2013, fig. 4a, 6 and 7c, respectively). The style with ornamented tip (Fig. 3F) appear e.g., in Clathria (Microciona) capverdensis or Clathria (Microciona) boavistae; both are noted from Cape Verde (van Soest, Beglinger & de Voogd, 2013, fig. 23 and 19, respectively). The straight style with ornamented head (Fig. 3G) is similar to Clathria (Microciona) strepsitoxa (compare with van Soest, Beglinger & de Voogd, 2013, fig. 15a1) noted from (among others) Alboran Sea today (de Voogd et al., 2023). The other acanthostyles (fig. 3H–J) could also have belonged to some Clathria or Antho species due to their shape and ornamented heads.

It is worth noting that the character and general morphology of the studied styles does not allow to assign them to any species with certainty. Moreover, the Clathria species in general, are very numerous in the areas adjacent to the study area.

Order Tethyida Morrow & Cárdenas, 2015

Family Tethyidae Gray, 1848

Genus Tethya de Lamarck, 1815

Type species. Tethya aurantium (Pallas, 1766) (type by subsequent designation).

Tethya sp.

Figures 3K and 3L

Material. Three spicules.

Description. There are several oxyasters with a diameter ranging from 60 to 130 µm, and with conical rays; some of the rays are diverging at the end.

Remarks. Such characteristic oxyasters can be found in many species of this genus, e.g., in Tethya wilhelma and T. minuta. These two species possess spicules with divided/split ray tips (compare with Sarà et al., 2001, fig. 6 and 15). T. wilhelma and T. minuta, despite both being described from an Aquarium of the Zoological-Botanical Garden “Wilhelma” in Stuttgart, Germany, are thought to have an Indo-Pacific origin (Sarà et al., 2001).

Order Tetractinellida Marshall, 1876

Family Theneidae Gray, 1872

Genus Annulastrella Maldonado, 2002

Type species. Annulastrella annulata Carter, 1880 (by monotypy).

Annulastrella cf. ornata (Sollas, 1888)

Figure 3M

Material. Two spicules.

Description. Very characteristic, annulate triods with uneven, 150 to 190 µm long rays with well-defined narrow, smooth annuli. There are about 20 to 25 annuli on each ray, but not all of them continue on the whole ray diameter. The annuli cover the whole length of the rays including the point of contact of the rays.

Remarks. The annulate triods resemble spicules of the theneid sponges Annulastrella ornata (Sollas, 1888) and A. annulata (Carter, 1880). The spicules of A. ornata are of comparable size, however they seem to be less annulated. Also A. verruculosa (Pulitzer-Finali, 1983), which is noted from the Mediterranean is similar in terms of shape, but smaller (rays up to 80 µm long). Today, Annulastrella ornata is known today from, among others, the western Mediterranean (Alboran Sea; de Voogd et al., 2023).

Family Thoosidae Cockerell, 1925

Genus Alectona Carter, 1879

Type species. Alectona millari Carter, 1879 (type by monotypy).

Alectona cf. millari Carter, 1879

Figure 3P

Material. Single spicule.

Description. Acanthoxea which is centrally bent and uniformly thinning to the spicule both ends. This 700-µm-long spicule possess minute spines regularly arranged along the whole length.

Remarks. This spicule resembles those of thoosid Alectona millari, however, the spicules of modern species are about two times smaller than the fossil one (Rützler, 2002b). This species is widely distributed all around the world (e.g., the Azores, Mediterranean Sea, including East European Atlantic Shelf; de Voogd et al., 2023); despite being mainly shallow-water species, several specimens of this sponge were found boring in deep-water corals around Madeira (Rützler, 2002b).

Family Thrombidae Sollas, 1888

Genus Thrombus Sollas, 1886

Type species. Thrombus abyssi (Carter, 1873) (type by subsequent designation).

Thrombus abyssi (Carter, 1873)

Figures 3N and 3O

Material. Two spicules.

Description. Short-shafted, spiny triaenes with clads divided trichotomously called acanthotrichotriaenes. The clads are about 30 µm in length divide on the first 10 µm of their length.

Remarks. The described acanthotrichotriaenes are of identical shape, ornamentation and size with those of the Recent species Thrombus abyssi Carter, 1873. Today, T. abyssi inhabits the Mediterranean and the Atlantic, including East European Atlantic Shelf (de Voogd et al., 2023).

Tetillidae indet.

Figures 3R and 3S

Material. Four spicules.

Description. There are several anatriaenes with long, thin clads measuring from 80 up to 200 µm. Their rhabds are up to 600 µm of length, but some of them are incomplete and their length may be in fact greater.

Remarks. These anatriaenes with characteristically curved long and thin cladomes could have belonged to sponges of the family Tetillidae. Although, anatriaenes are also characteristic for the family Ancorinidae. In the Gibraltar, there are many tetillid species recorded; among them are Craniella cranium and C. azorica (Boury-Esnault, Pansini & Uriz, 1994, fig. 31 and 33, respectively) but both are characterized by very long (up to 4 mm) anatriaenes. With only single complete and several incomplete spicules, the more precise assignment of these spicules as to Tetillidae is impossible.

Suborder Astrophorina

Astrophorina indet.

Figures 3U–3Y

Material. Several spicules.

Description. In the studied material there are several spicules that belong to suborder Astrophorina due to their tetraxial symmetry. Those are dichotriaenes (Figs. 3W and 3X). The first one is characterized by long, slender, dichotomously divided clads (that divide at 50 µm of length; the deuteroclads are at least 70 µm long but their exact length might be greater due to broken off tips). The other triaene is long-shafted orthomesotriaene which rhabd is 400 µm long and covered by small mammilliform outgrowths; the dichotomously divided clads are about 80 µm long (Fig. 3X). Another triaene recognized in the material is about 250 µm long, stout anatriaene with 50 µm long clads (Fig. 3Y). There are several calthrops in the studied material (e.g., Fig. 3V) which rays are 120 to 260 µm long and a single triod with about 220 µm long rays (Fig. 3U).

Remarks. The long triaenes (as well as anatriaenes assigned here to Tetillidae indet.) could have belonged to Geodia (family Geodiidae). However, lack of other spicule characteristic for this genus does not allow for the unquestionable assignment. Some of the described trienes could have belonged to tetillid sponges as well (e.g., Figs. 3W and 3Y). The assignment of triods and calthrop to a lower taxonomical level is not possible.

“Lithistida” indet.

Figures 3Z–3BB

Material. Three spicules.

Description. There are two irregular desmas of tetractinellid symmetry called tetraclones. The first tetraclone’s clads are about 100 µm long, smooth and strongly branching on their distal parts (Fig. 3AA). The other one is big, with one preserved clad of about 100 µm of length (the other two clads fragments are incomplete). The clads of this desma are rarely covered with wide and low projections and are branching on the tips (Fig. 3BB). The third desma is a rhizoclone, 220 × 170 µm, covered with numerous spines and outgrowths along the whole spicule (Fig. 3Z).

Remarks. The character of the studied spicules does not allow to assign them to any particular “lithistid” family, however, there is a high chance that the rhizoclone belonged to some sponge from the family Azoricidae or a rhizomorine “lithistid” (see Łukowiak et al., 2022a; Pisera & Lévi, 2002).

Class Hexactinellida Schulze, 1885

Subclass Hexasterophora Schulze, 1886

Order Lychniscosida Schrammen, 1903

Lychniscosida indet.

Figure 4A

Figure 4 Sponge spicules of the late Miocene of the Guadalquivir Basin (Hexactinellida).

(A) Fragment of lychniscosan skeleton; (B and C) Skeleton fragments of Tretopleura sp.; (D–F, J, K) pinnular penta- and hexactines of Hexactinellida indet.; (G–I) diactines of Hexactinellida indet.; (L) ornamented pentactine of Rossellidae indet.; (M, N, R) fragments of skeletons of Sceptrulophora indet.; (P) tyloscopule of Sceptrulophora indet. (transmitted light); (O) stauractine of Nodastrella cf. nodastrella; (S) discaster of Nodastrella cf. nodastrella; (T) hypodermal pentactine of ?Rossellidae indet.; (U–W) skeleton fragments of Sceptrulophora indet.

Material. Single spicule.

Description. A single fragment of lychniscosan skeleton which is characterized by a lantern-like structure that is created by fusion of lychniscosan hexactines. This dictyonal frame is about 100 µm in diameter and seem to come from the peripheral part of the framework as at least one of the “rays” seem to be originally preserved (unbroken) and measures about 100 µm of length.

Remarks. This skeleton fragment can belong to one of the two lychniscosan families, Aulocystidae Sollas, 1887 or Diapleuridae Ijima, 1927. Both of them are characterized by presence of such joints within the skeleton. There are not many species within these two families and among them only two, Neoaulocystis polae (Ijima, 1927) and Neoaulocystis grayi (Bowerbank, 1869) (both aulocystids) are noted from the nearby areas. The first one is recorded from the Red Sea and the second one from the eastern Caribbean (de Voogd et al., 2023). Thus, it is highly possible that this dictional skeleton fragment belongs to one of these species.

Genus Nodastrella Dohrmann et al., 2012

Type taxon. Nodastrella Dohrmann et al., 2012

Nodastrella cf. nodastrella

Figures 4S and 4O

Material. Two spicules.

Description. There is a single hexactinellid microsclere noted in the studied material. This is discaster—a spicule about 110 µm in diameter, with spherically arranged, ornamented primary rays with blunt tips (Fig. 4S). There is also other spicule which is ornamented dermal stauractine with 160-µm-long rays that might belong to the same taxon (Fig. 4O).

Remarks. Microscleres which are quite unique spicules in fossil record give a high chance to be assigned to species level. In this case, the presence of very characteristic discaster together with stauractine allows to assig these spicules to a rossellid species Nodastrella nodastrella which is characterized by these two spicule types in its skeleton (compare with Dohrmann et al., 2012, fig. 1b and fig. 2e). Moreover, there is a slender pentactine (Fig. 4Q) in the studied material that could have belonged to this species as well (compare with Topsent, 1915, fig. 2h). Likewise, it is possible that some of the diactines (Figs. 4G–4I) found in the fossil material could also have belonged to N. nodastrella. This species is today recorded from N Atlantic (Azores, Canaries, and Madeira) and Eastern Coast of the Atlantic Ocean (i.e., Florida; de Voogd et al., 2023).

Rossellidae Schulze, 1885

Rossellidae indet.

Figures 4L and 4T

Material. Two spicules.

Description. The first spicule is ornamented pentactine with 80-µm-long, swollen ray tips (Fig. 4L). The second one is slender hypodermal pentactine with rays which are at least 250 µm long (all the rays are broken) (Fig. 4T).

Remarks. The ornamented pentactine can belong to some rossellid species of Crateromorpha or Acanthascus. The slender hypodermal pentactines appear in many species of rossellids (Tabachnick, 2002). However, it also appears in all three amphidicosid families Hyalonematidae, Monorhaphididae and Pheronematidae, so the assignment to rossellids is here tentative.

Order Sceptrulophora Mehl, 1992

Sceptrulophora indet.

Figures 4M, 4N, 4P, 4R, 4U–4W.

Material. Two spicules and four skeleton fragments.

Description. First pentactine is characterized by actines located at one plane. They are about 170 µm long and delicately ornamented on the actins’ tips. They also possess very characteristic, better pronounced ornamentation on the upper surface of the actines. The opposite actine is incomplete (Fig. 4M). The other pentactine possess about 150 µm long actines. This spicule is ornamented along the whole spicule length, but the ornamentation is better pronounced on the actines’ tips. This spicule possesses a short knob on the apical part of the actine (not developed sixth actine; Fig. 4N).

There is also a single, fragmentarily preserved tyloscopule noted in the transmitted light (Fig. 4P). This very characteristic spicule is about 1,000 µm long and with well-developed, four ornamented rays with tylote tips.

Among the three skeleton fragments, the first is characterized by delicately swollen nodes. The nodes seem to be delicately ornamented (Fig. 4R). The other two skeletal fragments are fragments of dermal framework with irregular openings of epirhyses. The openings are from 10 × 20 µm to over 100 × 1,000 µm (Figs. 4U–4W).

Remarks. Such ornamented pentactines appear in many families among Sceptrulophora, e.g., in Farreidae, Uncinateridae, or Euretidae. Sponges of the family Farreidae are quite abundant in the Mediterranean Sea (de Voogd et al., 2023), especially genus Farrea (Reiswig, 2002a). There are three species of Farrea living in the nearby area, e.g., in the Saharan Upwelling zone (F. occa Bowerbank, 1862) and in the Azores, Canaries, and Madeira (F. foliascens Topsent, 1906 and F. laminaris Topsent, 1904 (Topsent, 1928, pl. 4, fig. 6 and 8, respectively)). Thus, these spicules might have belonged to one of them but the assignment is not unequivocal.

The pentactine with undeveloped sixth actine can be found in some sceptrulophoran families, e.g., Euretidae and Farreidae, but appear also in lyssacinosidan Rossellidae.

The skeleton fragment with swollen nodes could belong to some of the euretid sponges, but its more precise assignment is not possible.

Tyloscopules appear in all sceptrulophorid families except Cribrospongiidae. These fragmentarily preserved frameworks are similar to frameworks of many sceptrulophoran groups, but some euretid (see Reiswig & Wheeler, 2002) and aphorocallistid (Reiswig, 2002b), sponges inhabit nearby areas, so it could have belonged to a sponge of one of these two families. The euretid Gymnorete alicei, which inhabits Azores and the Saharan Upwelling zone (de Voogd et al., 2023) possess very similar skeleton (Fig. 7b in Reiswig & Wheeler, 2002). Moreover, in the studied material we have found tyloscopule (Fig. 4P) which characterizes this species as well (Reiswig & Wheeler, 2002). Despite these similarities, the unequivocal assignment of the studied skeleton fragments is not possible.

Family Uncinateridae Reiswig, 2002b

Genus Tretopleura Ijima, 1927

Type species. Tretopleura candelabrum Ijima, 1927 (type by monotypy).

Tretopleura sp.

Figures 4B and 4C

Material. Two skeleton fragments.

Description. The fragmentary preserved skeletons; one of them is with a quadrangular mesh of about 60 × 20 µm in diameter and very characteristic four-fingered claw outgrowth (Fig. 4B). The other fragment is preserved as long fragment of a mesh and a five-rayed, delicately ornamented claw outgrowth (Fig. 4C).

Remarks. Despite these skeletons with characteristic outgrows are preserved only fragmentarily, they seem to be identical with skeleton fragments of modern uncinaterid species Tretopleura candelabrum (compare with Ijima, 1927, pl. XXII, fig. 1). Today, this species which is the only known representative of this family, is recorded from the Banda Sea (de Voogd et al., 2023).

Hexactinellida indet.

Figures 4D–4K

Material. Seven spicules.

Description. There are three big, up to 1,000 µm long, diactines which apparently are reduced forms (by reduction of four actines) of hexactines. They are characterized by slightly curved axis and rounded actines tips. These spicules are sculptured with relief increasing on the actines’ tips (Figs. 4G–4I).

Other spicules are small, stout pinnular penta- and hexactines with about 50 µm long actines and a well-developed slender pinnule (Figs. 4D, 4E, 4J, 4K). These dermal spicules are also sculptured with the ornamentation increasing on their actines’ tips. Sometimes double pinnule hexactines appear (Fig. 4F).

Remarks. The big diactines resemble the spicules of sponges of the family Aphrocallistidae, especially genus Aphrocallistes. Likewise, the pinnular hexactines found in the fossil material are of comparable shape (but bigger) of pinules of some modern Aphrocallistes, e.g., A. beatrix Gray, 1858. This species is noted from the studied area (de Voogd et al., 2023; compare with Lopes, Hajdu & Reiswig, 2005, fig. 3). On the other hand, similar diactines can be found in other families e.g., Rossellidae, Leucopsacidae, Euplectellidae, or Hyalonematidae. The same is true for dermal pinules; they are recorded from families Euretidae, Pheronematidae, and Hyalonematidae (Reiswig & Wheeler, 2002; Tabachnick, 2002).

Discussion

Our study shows that the Atlantic part of the Atlanto-Mediterranean Seaway was inhabited by unexpectedly rich and diverse sponge fauna about 11.6–5.3 million years ago (Tortonian-Messinian). It comprised members of two sponge classes, Hexactinellida and Demospongiae, and included at least three “lithistids”. With respect to demosponges, we recognized at least thirty-four taxa. Of those, 17 could have been assigned to the species level and eight to the genus level. Hexactinellids were less common; still, they were represented by at least six taxa, including one that can be tentatively assigned to the species level and one that is assignable to the genus level.

Comparisons with other Miocene sponge faunas of Europe

The sponges described herein form the richest assemblage that has been reported from the Miocene of Europe to date. Several occurrences of dissociated sponge spicules have been reported from the European strata (for more details see Table S1). These include the sponge assemblage described by Pisera, Cachao & da Silva (2006) from Portugal that comprised “lithistids” (Rhizomorina), hadromerids (Alectona wallichii), spirophorids (Samus), astrophorids (Geodia and Erylus), and hexactinellids (Lychssacinosa). Of the taxa recognized in the material of Pisera, Cachao & da Silva (2006) only “lithistids” and hadromerids have been observed in our assemblage. In turn, the early Messinian sponge assemblage from the Turre Formation (Carboneras-Níjar basin, S Spain) described by Hoffmann et al. (2020), which was part of the northern edge of the Mediterranean basin at that time, includes three taxa. The demosponges from the Turre Formation are represented by triactinal tetraxons and calthrops, which were assigned by the authors to pachastrellids, whereas hexactinellids were represented by euretids and lychniscosans. While the pachastrellid spicules do not seem to have equivalents in our material, the hexactinellid skeleton fragment assigned by Hoffmann et al. (2020) to Euretidae shows resemblance to fragments found in our material. Lychniscosans was confirmed in both these assemblages as well. Also, the spicule assemblage from the Messinian of N Italy, despite recognizing 14 taxa (Costa et al., 2021), has not had much in common with our material. Only the spherasters of Tethya can be found in both these assemblages. This might be due to a shallower character of this Mediterranean assemblage.

Surprisingly, in terms of shared taxa, the studied assemblage exhibits the greatest similarity with the middle Miocene sponge fauna of Paratethys (Vienna Basin, Slovakia) described by Łukowiak, Pisera & Schlögl (2014). This assemblage was interpreted to have originated from deep waters and contained at least thirteen spicule types. Among them, at least five seem to be common for these two assemblages (those belonging to Anulastrella, Thrombus, Alectona, Monocrepidium, and Lychniscosida, as well as anthasters similar to those of Tethya).

In terms of bathymetry, the studied assemblage shows a mixture of species typical for shallow waters (e.g., Mycale (Mycale) grandis, Sceptrella biannulata, Placospongia decorticans) and species characteristic for waters of considerable depths (e.g., Hamacantha (H.) lundbecki, Histodermella ingolfi, Discorhabdella tuberosocapitata, Thrombus abyssi). This discordance in bathymetrical preferences among the recognized taxa might be explained either by mixed character of the assemblage (e.g., due to transportation), different bathymetric preferences of some of the recognized taxa in the past, or specific water conditions in the studied area. We are inclined towards the third option because the coastal upwelling that took place in this part of the basin (López García, 1995) has likely changed water conditions that enabled the taxa with deep-water preferences to migrate into neritic depths during the late Miocene.

There is a general trend that indicates a depletion of the Mediterranean sponge fauna as we approach the MSC. Interestingly, the sponge fauna that inhabited the corridor between the Mediterranean and the Atlantic Ocean was doing well before as well as close to the MSC, being very diverse and rich.

Biogeography

From among eighteen sponge species, eight inhabit the studied area to these days (Bubaris subtyla, Placospongia decorticans, Hamacantha (H.) lundbecki, Hamacantha (V.) papillata, Hamacantha (H.) johnsoni, Mycale (R.) marshallhalli, Alectona millari, Thrombus abyssi), four are recorded today from neighboring areas, such as the Western Mediterranean (Plocamione dirrhopalina, Monocrepidium vermiculatum) and the Alboran Sea (Crambe tuberosa, Annulastrella ornata), five are noted today from the Azores (Discorhabdella tuberosocapitata, Sceptrella biannulata, Nodastrella nodastrella) and Northern Atlantic (Histodermella ingolfi, Euchelipluma pristina), and one species is known from the Red Sea (Mycale (Mycale) grandis; for more details see Table S2).

Only four taxa seem to have no close relatives in adjacent waters and are possibly closer to sponges inhabiting distant parts of the world. In the case of spicules assigned to Acarnus sp., the morphologically closest morphotypes belong to Acarnus claudei—a species noted form S African coast (de Voogd et al., 2023). The oxyasters that we assigned to Tethya sp. also exhibit the greatest resemblance to spicules of distant, Indo-Pacific, species of Tethya (T. wilhelma and T. minuta). The same is true for hexactinellid skeletal fragments assigned to Tretopleura sp. The three known species belonging to this genus are currently recorded from the Banda Sea and W Pacific Ocean (de Voogd et al., 2023). Similarly, the spicule assigned here to Discorhabdella sp., despite showing some resemblance to the Mediterranean species, is most similar to those of D. hispida and D. misakiensis from Japan (Ise et al., 2021). This might indicate that the populations of species of Acarnus, Tretopleura, Tethya, and Discorhabdella had a wider geographic range in the past. This is the second case when spicules of the Alboran Sea species of Discorhabdella are more similar to geographically distant relatives than to species inhabiting neighboring areas (for more details see Boury-Esnault, Pansini & Uriz, 1992).

In turn, the contemporary geographical range of Mycale (Mycale) grandis covers the area of the Red Sea, East Africa coasts, Indonesian, and Pacific waters but not the E Atlantic (de Voogd et al., 2023). Its presence in East Atlantic, recognized during our study, extends the range of Mycale (Mycale) grandis to the majority of sea waters around the equator. Likewise, the modern distribution of Alectona millari, which is recorded from the N and E Atlantic Ocean through the Mediterranean, and up to Levantine Sea, must be also a Tethyan relic as this species (or spicules of this species) is recorded from the Miocene deposits of paleo-Mediterranean (Costa et al., 2021) and Eocene of Australia (Łukowiak, 2015).

The Tethyan origin of the modern distribution of some sponge taxa has already been postulated by many authors, e.g., Boury-Esnault, Pansini & Uriz (1992), and widely discussed by Łukowiak (2015), Łukowiak et al. (2022b). The studied assemblage provides new data to the discussion about the Tethyan relicts of the pre-Messinian biota.

Conclusions

We have recognized a rich and diverse sponge fauna from the Guadalquivir Basin, southwestern Spain, that has thrived before the onset of the Messinian Salinity Crisis. The community consisted of at least thirty-four taxa of demosponges and six hexactinellids.

From among eighteen taxa recognized to the species level, at least eight seem to be inhabiting this area to these days and many other of the studied sponges are recorded from adjacent areas, such as the Western Mediterranean, South European Atlantic Shelf, and the Azores. However, a group of sponge taxa identified in the studied sample were probably more closely connected to Indo-Pacific species, than to the species from the area. Such findings indicate that their geographical range has been wider in the geological past. The modern distribution of these taxa, which differs from their Miocene range, is an effect of a disruption after the closure of the Gibraltar strait. The shrinking ranges and disappearance from the Mediterranean Sea, resulting in disjunct modern distribution of some taxa, was likely the effect of the unfavorable conditions that took place in this area during the Messinian Salinity Crisis.

Supplemental Information

Supplemental Information 1 List of the other fossil occurrences of the recognized spicules.

Click here for additional data file.

Supplemental Information 2 List of taxa recognized to species level in the studied area and their closest to studied area of modern distribution and bathymetric preferences (from the literature data).

AS—Alboran Sea, WM—Western Mediterranean, AZ—Azores, NA—North Atlantic, RS—Red Sea, SEAS—South European Atlantic Shelf.

Click here for additional data file.

Thanks are given to Dr. Jesús Soria, for his advice and design of the geological map. We are also grateful to the owner of the Bodegas Fundador S.L.U. for permitting us to sample the vineyard area.

Additional Information and Declarations

Competing Interests

Author Contributions

Field Study Permissions

Data Availability

The authors declare that they have no competing interests.

Magdalena Łukowiak conceived and designed the experiments, performed the experiments, analyzed the data, prepared figures and/or tables, authored or reviewed drafts of the article, and approved the final draft.

Gerardo Meiro performed the experiments, authored or reviewed drafts of the article, gathered and photographed material, and approved the final draft.

Beltrán Peña performed the experiments, authored or reviewed drafts of the article, gathered material, and approved the final draft.

Perfecto Villanueva Guimerans performed the experiments, analyzed the data, authored or reviewed drafts of the article, and approved the final draft.

Hugo Corbí analyzed the data, prepared figures and/or tables, authored or reviewed drafts of the article, and approved the final draft.

The following information was supplied relating to field study approvals (i.e., approving body and any reference numbers):

Verbal permit of Rafael Rendon Gomez—general Director of Bodegas Fundador

The following information was supplied regarding data availability:

The material is housed in the collections of the Museo de Paleontología Universidad de Huelva, Spain, with the catalogue numbers AMUHU-CE-100 and the raw data are available in Figs. 2 and 3.

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
