# Peer review of "Miocene sponge assemblages in the face of the Messinian Salinity Crisis—new data from the Atlanto-Mediterranean seaway"

_PeerJ, doi:10.7717/peerj.16277_

## Round 0.1 · original submission · Minor Revisions

Dear Dr. Łukowiak,

You can see, that two reviewers proposed Minor revision and one - Major ones. Looking on the review of the second reviewer I propose you concentrate not only on his remarks noted in the text, but pay serious attention to the comments of the first reviewer concerning species identifications

Best regards,
Alexander

Reviewer 1 ·

Basic reporting

.

Experimental design

.

Validity of the findings

.

Additional comments

The interest of these new data on spicules from sponges of the Miocene sponge fauna of the Mediterranean is well presented. The authors interestingly underline the contribution of the study of fossil assemblages of loose spicules. Their material appears to be rather exceptional by the state of conservation of the spicules and the period near the Messinian Salinity Crisis in the Mediterranean Sea. The results clearly need to be published.
However, the suggested identifications, sometimes to the species level, are only suggestions based on a few free spicules compared to those of the present sponge fauna. This is of course normal in the conditions of the work, but I personally find that the tentative character of the identifications is in general not enough underlined, both in the descriptions and in the Discussion. Some points are discussed in my comments along the text.
The English language needs to be improved. The text also suffers of many misspellings and typing errors. Some, on technical terms which may escape to a professional editing service, have been noted with their line number. This clearly needs a thorough revision.
I also regret that the text and the illustrations are not in the same order, which is not easy to read. But I know that change would represent a significant amount of work.

Specific comments.

Lines 73 – 75 : are quite rare…. are scares in terms…. Scarce instead of scares, add and : and are scarce in term of….
157: For this species, as for most others, the identification from two spicules is tentative. It is based on other identifications from fossil spicules. The authors say “most likely”…
171: What are the differences of these “stout curved acanthostyles” with cricorhabds?
177-178: repetition
184, 199: what means “ornamented”? Slightly spinose, microspined?
185: thickness
191, 194: Rhabderemia, not Rhabdermia
318: The name of the species has a question mark. This could be more frequent. Or as Histodermella cf. ingolfi.
345: It seems to me that it is in Fig. 2 Q rather than P, that the fimbriae are not in the same plane
380: Acanthoxes are not found only in Histodermella.
418/ There is at last
421: enchanting acanthotylostyles?
435: why « pseudoastrose » ?
443: Schrammen
475: Here I think that this spicule, and possibly also V, W, may be reasonably attributed to Latrunculiidae.
500, 516 and several other places: nm long, instead of µm?
676-678: Style with ornamented tip (Fig. 3F): the presumed spines on the tip are not visible in Fig. 3F. Such styles are known in many other Demosponges. They could be referred with doubt to the genus Clathria; but the reference to the Indo-pacific C. vulpina, although with the mention e.g. , is not justified.
706: wilhelma
726: verruculosa
753: alectonid-like
866: pentactin
892: Sceptrulophora
906: epirhises?
910: Aspidoscopulia is not known from the Mediterranean Sea
960, 962: hecactinrs
980: tentatively assigned !

·

Basic reporting

The English is very good, but not quite fluent, with many awkward phrasings and misused or misplaced words (e.g. 'scares' for 'scarce', line 75); please have it edited by a fluent English speaker to make sure it reads clearly.
Illustrations are excellent.

Experimental design

Systematic Palaeontology. It would also be useful to have an introductory common on the philosophy behind assigning individual fossil spicules to extant taxa, with a perhaps-ten-million-year gap. I'm aware that at least some sponge groups have very stable, long-lived species, but for most groups of animals there would be an assumption of some evolutionary change and extinctions over this interval. If the taxa were not the same species, but rather a related taxon, then the assignments (often based on proximity to the ranges of living taxa) would be potentially misleading. I'm not suggesting that your approach is the wrong one, necessarily (it is probably the best one to use in order to provide palaeobiogeographic hypotheses), but for clarity I would like to see an explicit explanation of why you have approached the assignments in the way that you have.

Much rests on the assignment of the spicules to known taxa, which is made even when only a small number of spicules are available. This may be correct, but the justification is occasionally lacking. For example, the first species is justified only by “These spicules, most likely, belong to P. dirrhopalina...” Why is this the case? Even if such cricorhabds are known only from that genus, there are several other species. Is the likelihood being assessed by the morphology, or the range of the living species? Most taxa are much better justified than this one, but the basis for the assignment needs to be made clear in each case.

Minor points:
Geological Background: is there any evidence or published estimate of the water depth? If there are radiolarians, then some constraints should be possible, I believe? Also, are there any macrofossils in these beds, or trace fossils? Either would help to interpret the palaeoecological setting.

Line 152: authorship should normally be spelled out, rather than using 'et al.'

Line 880-884: do you really mean nanometres rather than micrometres? Also, the description of hypodermal pentactins doesn't seem to make these specific to Rossellidae, as such spicules also occur in, for example, the Amphidiscophora...

Validity of the findings

This really hinges on the reliability of the assignments (see above). If the approach were explained clearly at the beginning (i.e. the choice of assuming assignment to living taxa as the default, and partly choosing those taxa based on proximity to known ranges), and these assumptions are noted in the beginning of the Discussion section (together with implications of those choices, which I believe serve to emphasise potential similarities to the modern fauna, rather than differences), then I would be happy with the results being presented in this way.

One point to perhaps consider further is the anomalously wide range of water depths of the extant taxonomic assoignments. Mixing of assemblages (as mentioned) would be the obvious interpretation, if the depositional environment was deep. Where there are major contradictions, though, are related modern taxa limited to the same type of depth range, or could a different taxonomic assignment resolve the issue? More information would be useful here.

Additional comments

Overall, I do want to say how much O appreciate how well done this substantial work is, in general. There is a lot of material, and a lot of work has obviously gone into finding modern analogues for particular spicule types. This is important work, documenting the communities potentially represented by the spicule assemblages.

·

Basic reporting

Congratulations to this important scientific study. Lukoviak et al present a detailed taxonomic description of a highly diverse fossil sponge fauna from the Miocene, of Southern Spain. Miocene is a time from which so far not much has been published in terms of sponges. Furthermore, this fauna mainly contains sponges with soft skeletons, which are preserved only as isolated spicules. Contrary to sponges with rigid skeletons, these sponges are rarely found in the fossil record and their identification is a difficult challenge for the taxonomists. Here the authors document and describe 40 new taxa, 34 of the Demospongiae and 6 of the Hexactinellida, 18 of which are identified to species level. The paper is well-structured, citations seem complete and correct, the descriptions are documented by Photo plates (mainly SEM-photos of the sponge spicules) in high quality. The only real weakness I find in this MS is the English language, which is not professional, in many places incorrect or even not understandable. Especially bad is the English in the general chapters, such as Introduction (I bring a few examples below), Discussion. The Results, Systematic Paleontology part with the taxonomic descriptions are better, probably because these Descriptions follow a strict scheme and thus are easier to understand. There are a few misspellings and uncertainties as well. I strongly suggest that the authors work the text over once again and especially have this MS reviewed by a proficient English speaker, or professional editor, who has some knowledge on this topic, before it gets published. The conclusions are very clear and well documented by the results. I strongly recommend that this important paper gets published, as soon as the unclarities and linguistic points have been improved.

Experimental design

There are no real experiments in systematic paleontology, but the methods of investigations are well explained and up-to-date, e. g. the SEM-Photos are professional and high quality.

Validity of the findings

The main hypothesis that the Messinian Crisis did a heavy impact on the very diverse sponge fauna of the Pre-Mediterranean, and the present distribution of many sponge taxa is merely a relic of the much wider distributed fauna of the former Tethys Ocean. This is well documented and explained and furthermore documented by a table 1, showing the modern distribution and depth range of the 18 described fossil (to recent) species. This is a highly diverse and well-preserved, probably deep-water fauna of mainly soft-skeletal demosponges. Soft sponges are generally rarely preserved, more difficult to identify and less described compared to the rigid representatives. Also the Miocene is a period from which generally little is known or published on the sponges. However, it is important with respect to the impact of the Messinian crisis on the biodiversity of the Mediterranean marine fauna, especially concerning the sponges. This valuable, well-done work is important for paleontological and recent sponge science and for taxonomists and ecologist working on the Messinian, and Mediterranean biodivversity and (paleo)ecology..Please publish after some minor correction and rewriting.

Additional comments

Just a few examples of phrases that need some attention prior to publication, my comments are added within brackets:
Introduction:
62 ... incomplete. This is especially true in terms of studies of those taxa which tend to preserve (to be preserved!) as 63 disassociated skeletal elements
65 namely sponges of the class Hexactinellida and so called ìlithistidsî (an informal group (a polyphyletic group!) within the 66 demosponges),
67 by articulation. The sponge communities recreated from the assemblages of loose spicules are
68 relatively less frequent but very important as in modern seas the dominant group, the
69 demosponges, tend to preserve as dissociated spicules ...et al., 2022). This sentence is not understandable, please rewrite and improve!
139 Geological background ... The ratio of planktonic/benthonic 140 foraminifera range from 30%, and the benthic foraminifera are dominated by Cassidulinids,
141 Discorbids, Buliminids, Lagenids and Uvigerinids, which permit to interpret ... This sentence is not understandable, please rewrite and improve!
Many other phrases within this text need to be corrected and improved!
Furthermore:
Figure capture Fig 2: ; E. Wavy diactne; F. Wavy diactne of Bubaris subtyl (should be wavy diactine!)
Figure 4: M, N, R. Fragments of skeletons of Sceptrulophora indet.; I think these can be identified only as Hexactinosida indet. For Scepptrulophora we are missing the proof: sceptrules!
Figure 4; U 3W. Skeleton fragments of Hexactinellida indet.I think, these can be identified as Hexactinosida indet. because of the rigid skeleton (fragments) which looks quite hexactinosidan!

---

## Round 0.2 · accepted · Accept

Dear Dr. Łukowiak,

I have assessed the revision myself, and I am happy with the current version. Thank you for the detailed responses to the comments of the reviewers and for the corrections made to the manuscript.

Now your manuscript is ready for publication.

Best regards,
Alexander